# Using geographically weighted regression analysis to cluster under-nutrition and its predictors among under-five children in Ethiopia: Evidence from demographic and health survey

Amare Muche[1], Mequannent Sharew Melaku[2]*, Erkihun Tadesse Amsalu[1], Metadel Adane[3]

1 Department of Epidemiology and Biostatistics, School of Public Health, College of Medicine and Health Science, Wollo University, Dessie, Ethiopia, 2 Department of Health Informatics, Institute of Public Health, College of Medicine and Health Science, University of Gondar, Gondar, Ethiopia, 3 Department of Environmental Health, College of Medicine and Health Sciences, Wollo University, Dessie, Ethiopia

* mequsharew8@gmail.com

**Data Availability Statement:** All relevant data are available from the Demographic Health Surveys Program (https://dhsprogram.com/).

## Abstract

### Background

Malnutrition among under-five children is a common public health problem and it is one of the main cause for the mortality of under-five children in developing countries, including Ethiopia. Therefore, lack of evidence about geographic heterogeneity and predictors of under-nutrition hinders for evidence-based decision-making process for the prevention and control programs of under-nutrition in Ethiopia. Thus, this study aimed to address this gap.

### Methods

The data were obtained from the Ethiopian Demographic and Health Survey (EDHS) 2016. A total of 9,384 under-five children nested in 645 clusters were included with a stratified two-stage cluster sampling. ArcGIS version 10.5 software was used for global, local and ordinary least square analysis and mapping. The spatial autocorrelation (Global Moran's I) statistic was held in order to assess the pattern of wasting, stunting, and underweight whether it was dispersed, clustered, or randomly distributed. In addition, a Bernoulli model was used to analyze the purely spatial cluster detection of under-nutrition indicators through SaTScan version 9.6 software. Geographically weighted regression (GWR) version 4.0 software was used to model spatial relationships in the GWR analysis. Finally, a statistical decision was made at $p$-value<0.05 with 95%CI for ordinary least square analysis and geographically weighted regression.

### Main findings

Childhood under-nutrition showed geographical variations at zonal levels in Ethiopia. Accordingly, Somali region (Afder, Gode, Korahe, Warder Zones), Afar region (Zone 2),

**Funding:** The author(s) received no specific funding for this work.

**Competing interests:** The authors have declared that no competing interests exist.

**Abbreviations:** DHS, Demographic and Health Survey; EDHS, Ethiopian Demographic and Health Survey; HAZ, Height for Age Z-score; ICF, International Classification of Functioning, Disability, and Health; IQR, Inter Quartile Range; LLR, log-likelihood ratio; SD, Standard Deviation; UNICEF, United Nation International Children's Fund; WAZ, Weight for age Z-score; WHO, World Health Organization; WHZ, Weight for Height Z-score; EFY, Ethiopian Fiscal Year; FMOH, Federal Ministry of Health; GWR, Geographic Weighted Regression; OLS, Ordinary Least Square.

Tigray region (Southern Zone), and Amhara region (Waghmira Zones) for wasting, Amhara region (West Gojam, Awi, South Gondar, and Waghmira Zones) for stunting and Amhara region (South Wollo, North Wollo, Awi, South Gondar, and Waghmira zones), Afar region (Zone 2), Tigray region (Eastern Zone, North Western Zone, Central Zone, Southern Zone, and Mekele Special Zones), and Benshangul region (Metekel and Assosa Zones) for underweight were detected as hot spot (high risk) regions. In GWR analysis, had unimproved toilet facility for stunting, wasting and underweight, father had primary education for stunting and wasting, father had secondary education for stunting and underweight, mothers age 35–49 years for wasting and underweight, having female children for stunting, having children eight and above for wasting, and mother had primary education for underweight were significant predictors at ($p$<0.001).

## Conclusions

Our study showed that the spatial distribution of under-nutrition was clustered and high-risk areas were identified in all forms of under-nutrition indicators. Predictors of under-nutrition were identified in all forms of under-nutrition indicators. Thus, geographic-based nutritional interventions mainly mobilizing additional resources could be held to reduce the burden of childhood under-nutrition in hot spot areas. In addition, improving sanitation and hygiene practice, improving the life style of the community, and promotion of parent education in the identified hot spot zones for under-nutrition should be more emphasized.

## Background

Globally, under-five children are the most vulnerable segment of the population [1]. Under nutrition is one of the most public health burdens in developing countries. It includes stunting, wasting, and under-weight [2] and is determined through measurements of height, weight, and age [3,4].

Stunting has been defined as a child who is too short for their age [5]. Globally, stunting affected about 149 million under-five children in 2018. Of these 55% of stunted children lived in Asia and 39% lived in Africa [6]. Currently, in Ethiopia, the prevalence is estimated to be about 38% among under-five children [7].

Wasting has been also defined as a child who is too thin for their height. In 2018, worldwide wasting contributed to threaten the lives of an estimated 49 million under-five children. Of these, more than two-thirds of all wasted children are found in Asia and more than one quarter found in Africa [6]. The latest recent study in Ethiopia showed that the prevalence is estimated to be about 10% among under-five children [7].

Similarly, under-weight refers a child who is too small for his/her age which is the weight for age < -2 standard deviation(SD) of the WHO Child Growth Standards median [4,8]. Consecutively, the recent report in Ethiopia indicated that 'the prevalence is estimated to be about 24% among under-five children. The results of impaired growth and development in children can be life long and reduce academic performance and the ability to contribute to the nation [5]. Many factors contribute to child under-nutrition including inadequate diet and poor infant and young child feeding (IYCF) practices, a high burden of infectious disease, and a lack of basic infrastructure to enabling access to clean water, sanitation, and health services [9–11].

Different studies have been conducted to explore the spatial distributions of communicable diseases [12–20]. The application of spatial analysis to non-communicable diseases has become a common practice to date [21–23]. Consecutively, previous studies revealed that spatial variation of under-nutrition among under-five children [24–26]. Similarly, studies conducted in Ethiopia have also showed the spatial variation of childhood stunting [27–29]. However, spatial studies on under-nutrition status of children are limited and lack modeling of spatial relationships between the identified clusters of under-nutrition and its predictors.

Thus, understanding the area-based heterogeneity and factors affecting under-nutrition is a footstep for evidence-based decision-making in under-nutrition prevention and control programs. In addition, detecting spatial variation is also useful to recognize gaps in the performance of program on childhood nutrition that could not be identified through the routine monitoring of the nutritional status of children. Hence, this study aimed to explore the geographical variation of under-nutrition and its predictors among under-five children in Ethiopia using geographically weighted regression analysis.

## Methods

### Study setting

Ethiopia is found in the horn of Africa covering 1,104,300 km$^2$ and ranks 10$^{th}$ in Africa in land coverage. Ethiopia is a country with a great geographical diversity ranging from peaks up to 4,550 m above sea level down to a depression of 110 m below sea level. Ethiopia has nine administrative regions (Tigray, Afar, Amhara, Oromia, Somalia, Benishangul Gumuz, Gambella, Somalia, Harari and SNNPR) and two city administrations namely Diredawa and Addis-Ababa (Fig 1). The country is divided into 68 zones, 817 districts, and 16,253 *kebeles* based on the report of 2010 Ethiopian fiscal year. Contextually, it is categorized as agrarian, pastoralists and city-based population. It has a total of 104,957,000 populations, of which 36,296,657 were women. Majority of the population about, 83.6% living in rural areas and 16.7% of the population reside in urban areas. The average household size in national level is 4.7 persons [30]. The country has fertility rate of 4.6, infant mortality rate (per 1,000 live births) of 48, and child mortality rate (per 1,000 live births) of 67 children deaths. We used the 2016 EDHS data for this study. The EDHS waive was conducted from January 18 2016 to June 27, 2016 [7].

### Population

All children aged 0 to 59 months living in Ethiopia were considered as a source population and the study population includes all under-five children in the selected Enumeration areas during the EDHS data collection. A total of 9,384 children aged 0 to 59 months who full fill the inclusion criteria were considered for the entire analysis.

To select study participants, a stratified, two-stage cluster sampling technique was employed for the 2016 EDHS. Enumeration areas (EAs) were the sampling units for the first stage and households were the sampling unit for the second stage. In the 2016 EDHS, a total of 645 EAs (202 urban and 443 rural) were selected with a probability proportional to EAs size (based on the 2007 housing and population census) and independent selection in each sampling stratum. Of these, 18,008 households and 16,583 eligible women were included. The detailed sampling procedure was presented in the full EDHS report [7].

The 2016 EDHS spatial data (latitude and longitude coordinates) was used for this study. It was taken from the selected enumeration areas during the data collection period. From a total of 645 clusters included in the 2016 EDHS, 23 of them had no latitude and longitude coordinates. The location data were accessed through the web page of the measure DHS Program after being authorized for utilization of the data (Fig 2).

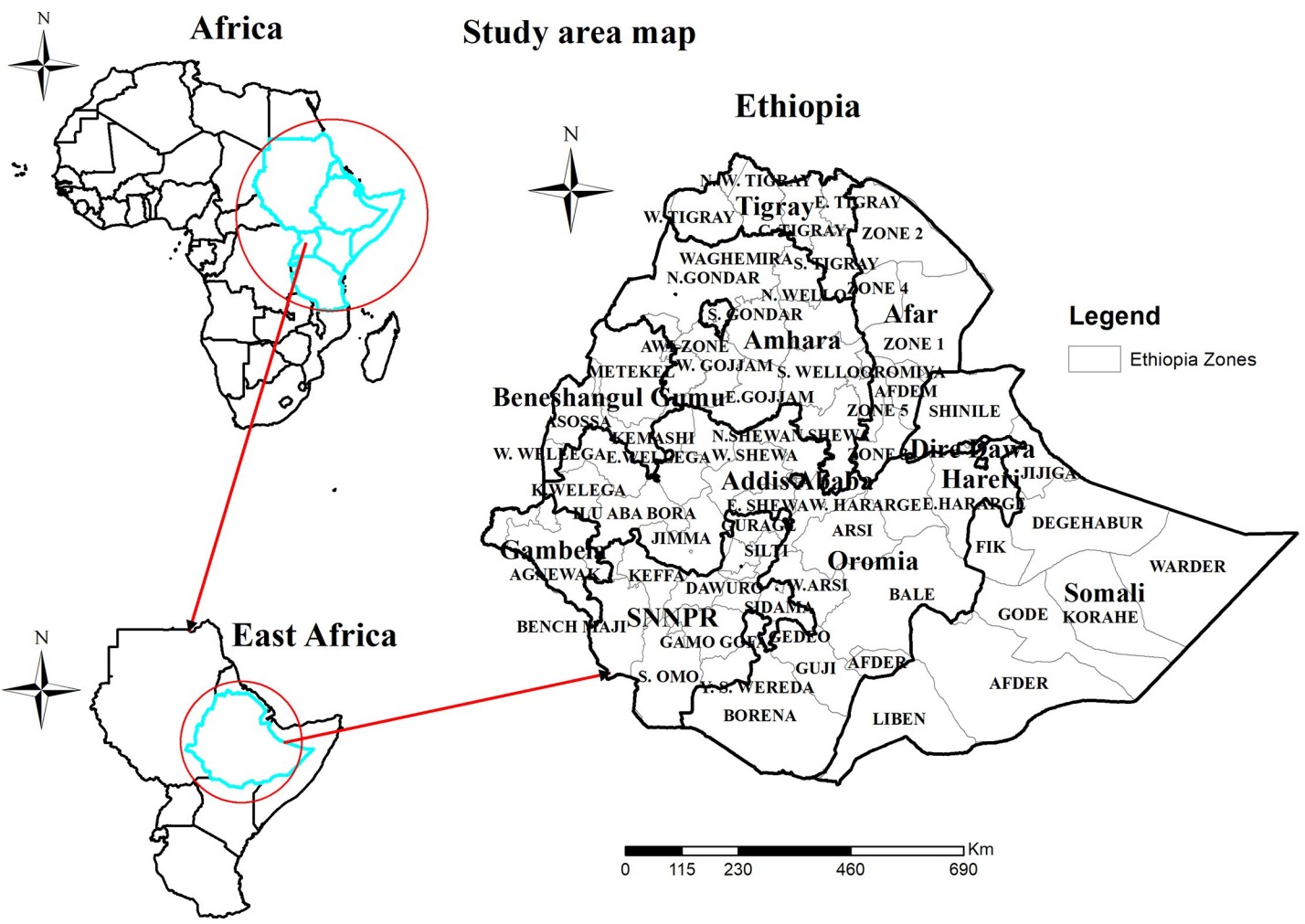

**Fig 1. Map of the study area.**

## Variables of the study

The outcome variable used for this study was under- nutrition which includes wasting, stunting, and underweight. Independent variables of the study include mothers' age, mothers' educational level, fathers' educational level, marital status, sex of children, type of toilet facility, drinking water source, distance from the health facility, residence, wealth index, and family size. Categorization of variables like education level of respondents, educational level of fathers, and family size were based on different literatures [31–34].

## Operational definition

**Under-nutrition.**   Defined as the type of malnutrition which includes stunting, wasting, and underweight among under-five children [1].

**Underweight.**   Defined as children who have <-2 SDs below the mean weight for age of the National Center for Health Statistics and the World Health Organization (WHO) reference population [1].

**Stunting.**   Defined as children who have low height/length for age Z score <-2 SDs of the median value of the WHO Child Growth Standards median aged 6–59 months [1,35].

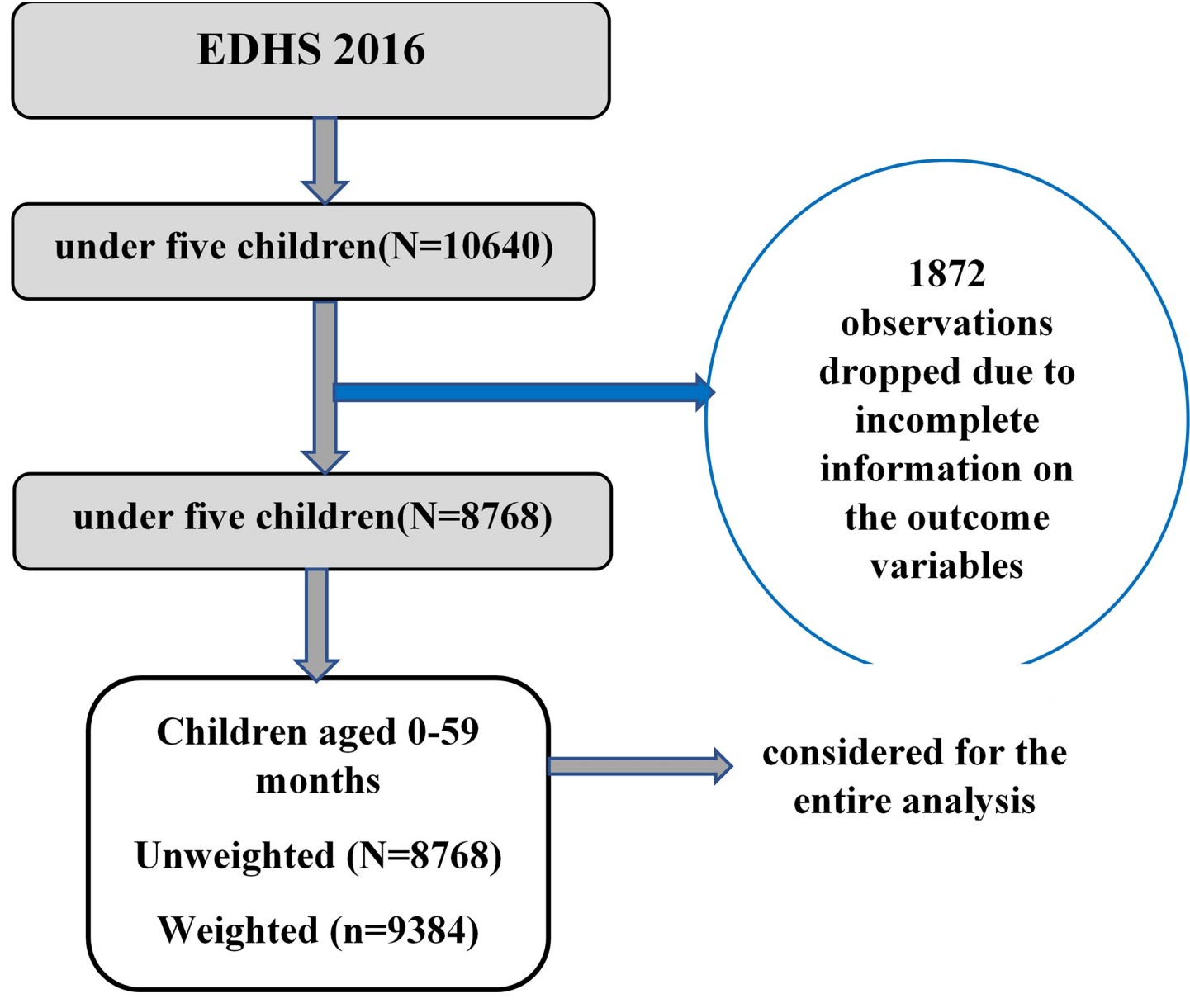

**Fig 2. Sampling procedure of the study.**

**Wasting.** Defined as the children who have low height/length-for-age-Z score <-2 SDs of the median value of the WHO Child Growth Standards median aged 6–59 months [1,35].

**Improved toilet facility.** A toilet facility includes any non-shared toilet of those types: flush/pour flush toilets to piped water systems, septic tanks and, and pit latrines; ventilated improved pit (VIP) latrines, pit latrines with slabs, and composting toilets [36].

**Unimproved toilet.** A type of toilet facility that includes a bucket or a toilet that flush's to elsewhere (in or nearby the household environment), pit latrine without slab/open pit, no facility/bush/field, bucket toilet, hanging toilet/latrine, no facilities [36].

**Improved source of drinking water.** Drinking water includes water piped into the residence, from piped water, a public tap, standpipes, tube wells, water from a borehole, a protected wells and spring, rainwater, and bottled water [36].

**Unimproved source of drinking water.** Drinking water from unprotected wells or springs, water from a vendor or tanker-truck and surface water (including rivers, dams, lakes, ponds, streams, canals, and irrigation channels [36].

**Anthropometric measurements of weight and height.** These measurements during the EDHS 2016 were taken with standardized and calibrated measuring tools after oral consent was obtained from their mothers. Accordingly, weight and height measurement of children was carried out from the selected enumeration areas. During data collection children less than 24 months were measured for height lying down and children greater than 24 months were measured while standing using a short measuring board. Weight measurements were taken with lightweight SECA mother infant scales with a digital screen designed and measured under the guidance of UNICEF. Details on anthropometric measurement were found on the EDHS report [7].

## Data management and analysis

After accessing the data from the MEASURE DHS website data extraction, data weighting, data cleaning, recoding, Descriptive and summary statistics were held using STATA version 14.1 software.

## Spatial analysis

The spatial autocorrelation (Global Moran's I) statistic was held in order to assess the pattern of wasting, stunting, and underweight whether it was dispersed, clustered, or randomly distributed in the study area. Details about spatial autocorrelation is published everywhere [37,38]. Local Moran's I measure positively correlated (high-high and low-low) clusters and outliers. The statistically determination of cluster outlier is published everywhere [39,40].

Gettis-ord Gi* statistics had been calculated to measure how spatial autocorrelation differs through the study location by computed Gi* statistics for each area. Z-score was calculated to ensure the statistical significance of clustering and the p-value calculated the significance $p$-value$<0.05$ at 95% CI if the Z-score is between -1.96 and +1.96, the $p$-value must be greater than 0.05 and vice versa. If the $p$-value is less than -1.96 it is declared as a cold spot and if greater than +1.96 it is declared as hotspot areas [41,42].

## Spatial scan statistical analysis

It tests the presence of statistically significant spatial clusters of wasting, stunting, and underweight among under-five children using Kuldorff's SaTScan version 9.6 software. Children who had been wasted, stunted, and underweight were considered as cases and children who had normal nutritional status as controls to fit the Bernoulli model [43,44].

Spatial cluster size $< 25\%$ of the population was used, as a higher boundary, which allowed both small and large clusters detection. The primary and secondary clusters were identified and assigned $p$-values and ranked based on their log likelihood ratio test [43–45].

## Spatial regression analysis

Spatial regression has both local and global analysis techniques [46–48]. Therefore, first, we had handled global geographical regression models and then local geographical analysis in order to ensure the variability of coefficients across each enumeration areas [49–51]. Then we have checked the assumptions using exploratory regression with the respective tests. The normality assumption was checked for residuals using Jarque-Bera test. As residuals are not spatially auto-correlated, confirming the koenker Bp test was done to check if the model under

gone for geographically weighted regression or not. Geographically weighted regression was executed using GWR version 4.0 software. We had also checked the six checks which recommended for a model undergone for spatial regression [52,53]. These are coefficients have the expected sign, no redundancy among model explanatory variables, coefficients are statically significant, strong adjusted $R^2$ values and the above two conditions stated before. Variables that have a *p*-value less than 0.05 are selected and described based on their coefficients.

## Ethical consideration

Permission to use the dataset has been granted by the Measure DHS program through legal registration. EDHS (2016) data was used which is available on the public domain through the Measure DHS website (www.measuredhs.com). Accordingly, the investigators had requested permission to use the data set on the Measure DHS website about the geographical variation of wasting, stunting, and underweight among under-five children in Ethiopia in May, 2020.

## Results

### Socio- demographic and other characteristics of respondents

A total of 10,640 under-five children were included in the 2016 EDHS survey. Of these, 1872 under-five children were dropped since valid or complete data were not obtained during data collection. Thus, after weighting the data a total of 9384 under-5 children were included in the analysis. The median age of children were 28 months with IQR = 31. In addition, the mean height of mothers was 158 centimeters (cm) with SD = 6.7. About 6183 (65.9%) of mothers had no formal education and 8363 (89.1%) were rural residents. About, 8471(89.9%), of respondents had used improved drinking water and about, 5278(56.25%) of respondents had used unimproved toilet facility (Table 1).

### Nutritional status

The prevalence of stunting, wasting and under-weight among under-five children in Ethiopia was 3598 (38.30%) (95% CI: 37.34–39.30), 949(10.10%) (95% CI: 9.51–10.73) and 2192 (23.54%) (95%CI: 22.70–24.40), respectively. The study showed variation of under- nutrition across regions of Ethiopia. Consecutively, stunting was lowest in Addis Ababa (15%) and highest in Amhara region (46%), wasting is lowest in Addis Ababa (3.4%) and highest in Somali region (20.9%) (46%), and under-weight is lowest in Addis (4.4) and highest in Afar (37.8) (Fig 3).

### Spatial variation of under-nutrition in Ethiopia

The spatial distribution of under-nutrition (wasting, stunting, and under-weight) was clustered at zonal level. Hence, the global Morans I index value was 0.363 (*p*-value < 0.001), 1.072 (*p*-value <0.001) and 0.879 (*p*-value <0.001) for wasting, stunting, and under-weight respectively as shown below in Table 2 and Fig 4.

The highest case distribution of stunting was spatially clustered in the Amhara region (North Gondar, South Gondar, Waghmira, and East Gojam zones), Afar region (Zone 1), and Oromia region (Guji, Borena, and West Arsi zones). Zone 4 of Afar region, Afder, Liben, Shinale, Degahbur, and Warder Zones of Somali region, Neuer zone of Gambella region was highest in wasting children case distribution. The highest case distribution of under-weight was spatially clustered in the Amhara region (Waghmira and North Wollo zones), Afar region (Zone 2 and Zone 3), and Somali region (Liben zone) (Fig 5).

Table 1. Socio-demographic characteristics of the participants in Ethiopia (*N* = 9384).

| Variables | Frequency (n) | Percentage (%) |
|---|---|---|
| Age the child (in months) | | |
| <6 | 1013 | 10.9 |
| 6–11 | 1013 | 10.9 |
| 12–23 | 1853 | 19.75 |
| 24–35 | 1766 | 18.82 |
| 36–47 | 1814 | 19.33 |
| 48–59 | 1925 | 20.51 |
| Sex of the child | | |
| Male | 4801 | 51.2 |
| Female | 4583 | 48.8 |
| Age of the mother (in years) | | |
| 15–24 | 308 | 3.3 |
| 25–34 | 6778 | 72.2 |
| 35–49 | 2298 | 24.5 |
| Mother's educational level | | |
| No education | 6183 | 65.9 |
| Primary education | 2562 | 27.3 |
| Secondary education/above | 639 | 6.8 |
| Mother's employment status | | |
| Employed | 4101 | 43.7 |
| Not employed | 5283 | 56.3 |
| Residence | | |
| Urban | 1021 | 10.9 |
| Rural | 8363 | 89.1 |
| Region | | |
| Tigray | 642 | 6.84 |
| Afar | 90 | 0.95 |
| Amhara | 1807 | 19.26 |
| Oromo | 4137 | 44.1 |
| Somali | 393 | 4.2 |
| Benishangul-Gumuz | 100 | 1.1 |
| SNNPR | 1934 | 20.6 |
| Gambela | 22 | 0.23 |
| Harari | 19 | 0.21 |
| Dire Dawa | 36 | 0.38 |
| Addis Ababa | 205 | 2.2 |
| **Family size** | | |
| 2–4 | 2416 | 25.74 |
| 5–7 | 4679 | 49.86 |
| ≥8 | 2289 | 24.4 |
| Wealth status | | |
| Poorest | 2176 | 23.2 |
| Poorer | 2208 | 23.53 |
| Middle | 1980 | 21 |
| Richer | 1692 | 18 |
| Richest | 1328 | 14.2 |
| Source of drinking water | | |

(*Continued*)

**Table 1.** (Continued)

| Variables | Frequency (n) | Percentage (%) |
|---|---|---|
| Improved | 5278 | 56.25 |
| Unimproved | 4106 | 43.75 |
| Type of toilet facility | | |
| Improved | 913 | 10.1 |
| Unimproved | 8471 | 89.9 |
| Father's educational level | | |
| No education | 4291 | 48.14 |
| Primary | 3570 | 40.05 |
| Secondary/above | 1053 | 11.81 |
| Father's employment status | | |
| Employed | 8296 | 88.4 |
| Not employed | 1088 | 11.6 |
| Distance to health facility | | |
| Big problem | 5691 | 60.65 |
| Not big problem | 3693 | 39.35 |

## Cluster and outlier zone detection for under-nutrition

Local Moran's I analysis result of wasting, stunting, and underweight revealed that there were significant outliers. Accordingly, high outliers for underweight were detected in Shinale and Liben zones of the Somali region, Zone 3 of Afar region. Similarly, Shinale and Liben zones of the Somali region, Zone 3 of the Afar region, Guji, and West Shewa of the Oromia regions were detected as high outliers for wasting (Fig 6).

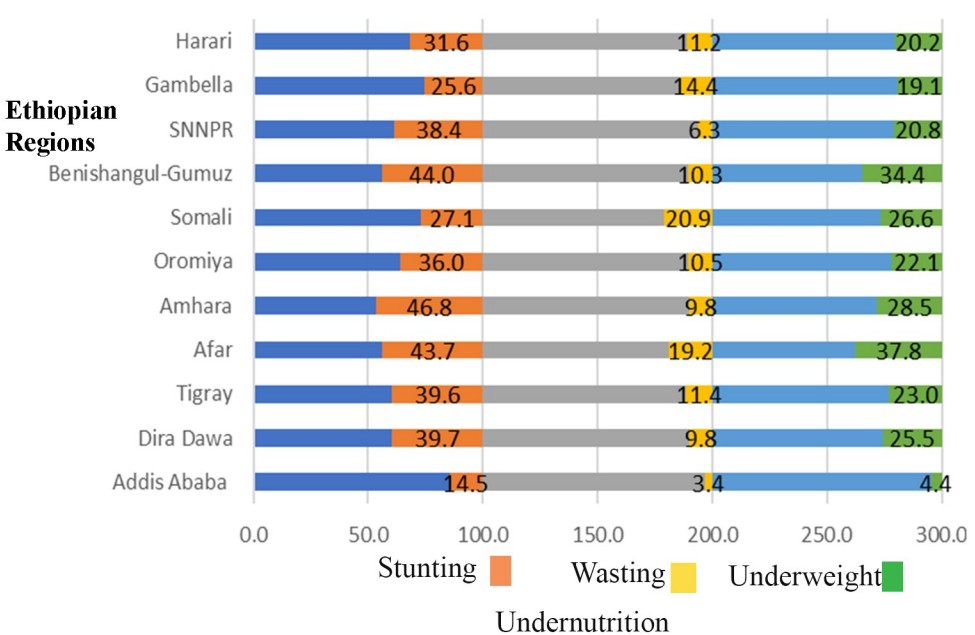

**Fig 3. Regional variation of under-nutrition among under-five children in Ethiopia.**

**Table 2. Summary of spatial autocorrelation of incomplete immunization in Ethiopia, EDHS 2016.**

| Variable | Observed moran's I | Expected moran's I | Z_score | *p*_value |
|----------|--------------------|--------------------|---------|-----------|
| Wasting | 0.362912 | -0.011494 | 4.157 | <0.001 |
| Stunting | 1.07198 | -0.01802 | 12.014 | <0.001 |
| Underweight | 0.87962 | -0.01495 | 9.889 | <0.001 |

## Hot spot and cold spot zones for under-nutrition in Ethiopia

We conducted a hot spot analysis using Gettis-Ord Gi* statistics. GIZ-score is computed to determine the statistical significance of clustering, and the *p*-value is computed for the significance. When the Z-score increases (+/−), its significance level increases. Statistical output with high Gi* indicates "hotspot" whereas low Gi* means a "cold spot". As shown in Fig 7, dark red colors show significant (*p*-value < 0.001) clusters of Under-nutrition (risk areas), whereas, dark blue colors show significant (*p*-value < 0.001) non-risk areas. The more clustered red and blue colors indicate more Under-nutrition risk and non-risk areas, respectively.

Hotspot analysis enables the detection of both extremities either high or low wasting, stunting, and underweight coverage zones.

Accordingly, hot spot (high risk) regions for stunting were detected in the Amhara region (West Gojam, Awi, South Gondar, and Waghmira zones). Somali region (Afder, Gode, Korahe, Warder Zones), Afar region (Zone 2), Tigray region (Southern zone), and Amhara region (Waghmira zones) was detected as hot spot zones for wasting.

Amhara region (South Wollo, North Wollo, Awi, South Gondar, and Waghmira zones), Afar region (Zone 2), Tigray region (Eastern zone, North Western zone, Central zone, Southern zone, and Mekele special Zones), and Benshangul region (Metekel and Assosa Zones) were hot spot zones for underweight.

On the other hand, North Shoa Zones of the Amhara region, Zone 3 of Afar region, Arsi and East Hararge of Oromia region, Shinale and Jijiga zones of the Somali region Dire Dawa city administration, Harari city administration cold spot zones for stunting.

Arsi, East Shoa, and South West Shewa Zones of the Oromia region, zone 3 Zone of Afar region, Gurage and Silte Zones of SNNPR for wasting were considered as cold spot zones.

Amhara region (North Shoa Zone), Afar region (Zone 3), Oromya region (Arsi, Horo Gudo Wellega, North Shoa, East Shoa, West Shoa, and South West Shoa Zones), SNNPR (Gurage and Silte Zones), and Addis Ababa city administration were detected as cold spot areas for childhood underweight (Fig 7).

## Spatial clustering of under- nutrition in Ethiopia

Spatial scan statistics identified primary and secondary clusters of wasting, stunting, and underweight using the maximum spatial circular windows ≤ 25% of the total population. Accordingly, spatial scan statistics identified (one primary and one secondary cluster) for stunting, (one primary and three secondary clusters) for wasting, and (one primary and three secondary clusters) for under-weight.

The primary cluster for stunting (LLR = 466.3, P<0.001) was centered at (14.033876N, 37.105923 E) with 578.81 km radius and a relative risk (RR) of 1.44. It incorporates all zones of Amhara, Tigray, Afar, and Benishangul regions. Kelem Wollega, Horo Gudu Wollega, West Shoa, South West Shoa, and North Shoa Zones of the Oromia region were also primary clusters for stunting.

The primary cluster for wasting (LLR = 215.9, P<0.001) was centered at (7.650693 N, 47.007919 E) with 819.42 km radius and a relative risk (RR) of 1.8. It surrounds all zones of

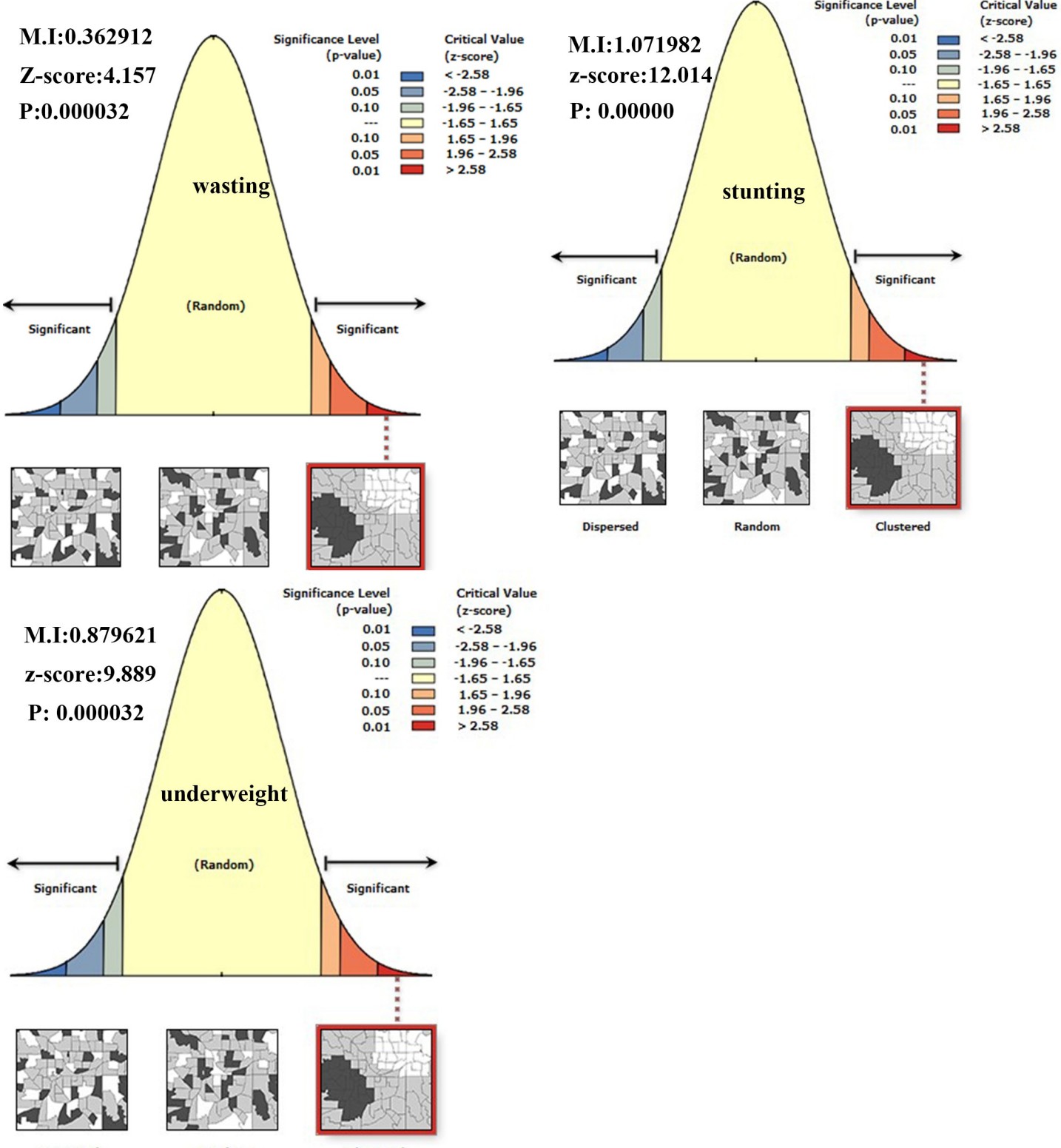

**Fig 4. Spatial autocorrelation of under-nutrition among under-five children in Ethiopia.**

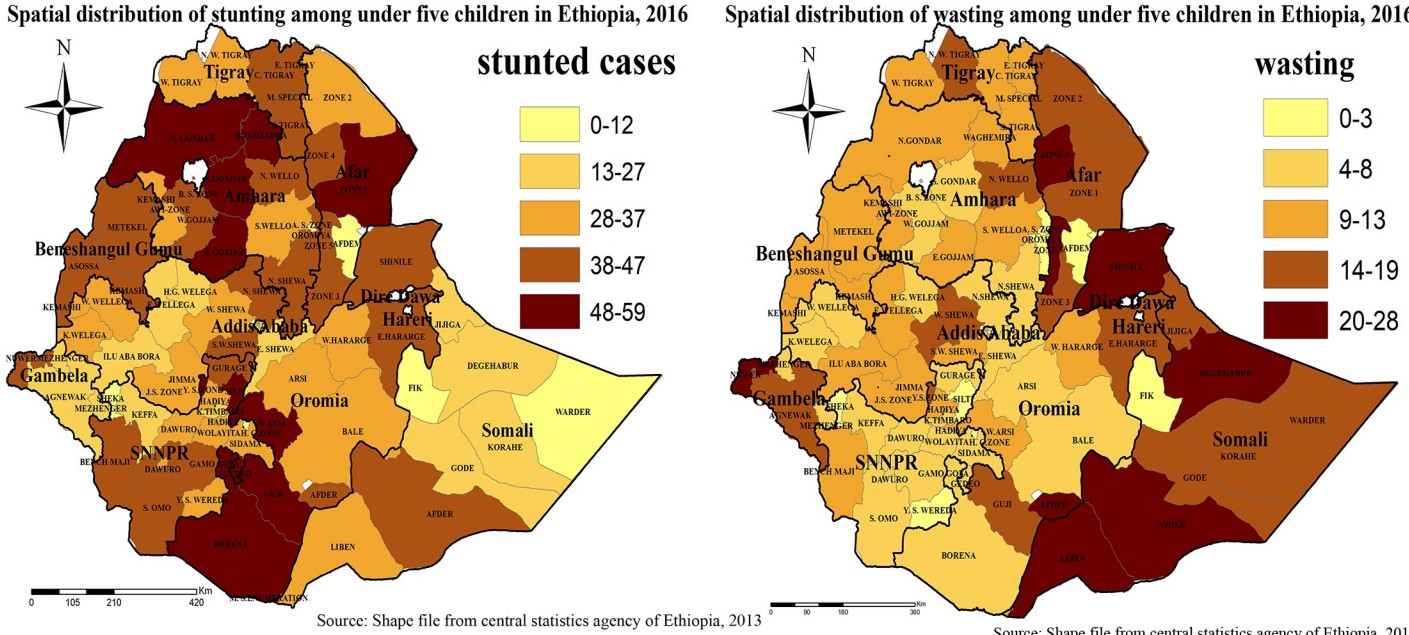

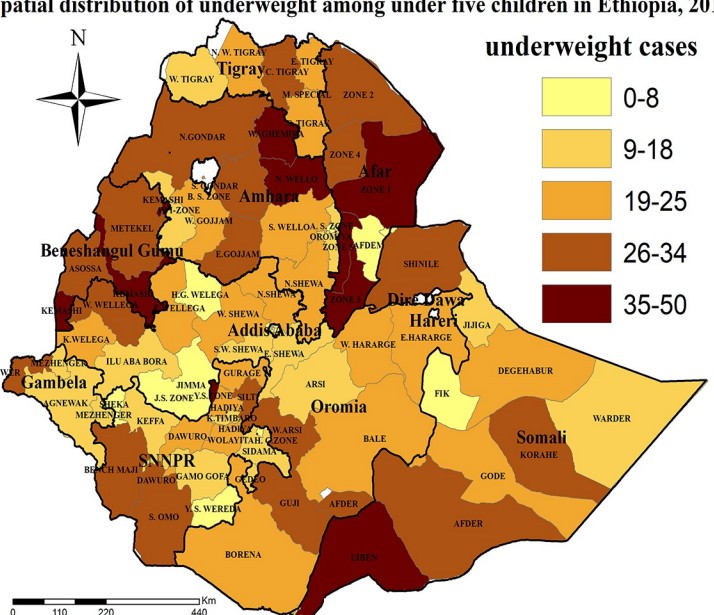

**Fig 5. Spatial variation of under-nutrition among under-five children in Ethiopia.** Spatial distribution of stunting, wasting and underweight in Ethiopia from Ethiopian demographic and health survey 2016. The right panel represents for stunting, the left panel represents for wasting the right bottom panel represents for underweight. In the figure the damp gray color indicates very high clustering, the bright gray color indicates high clustering of wasting, stunting, and underweight cases, the damp yellow color indicates moderate clustering, the bright yellow color indicates low clustering, the whitish yellow color indicates very low clustering of wasting, stunting, and underweight cases. The thick black line indicates reginal borders while the thin white black line indicates zonal borders. *Shape file source: (CSA, 2013; https:// africaopendata.org/dataset/ethiopia-shapefiles); Own Map output: using ArcGIS 10.7 Software analysis.*

Somali region, (Bale, East Harerege, West Harerege, and Arsi) Zones of Oromia region, and (Zone 1, Zone 3, and Zone 5) of Afar region.

The primary cluster for under-weight (LLR = 402.9, P<0.001) was centered at 14.033876N, 37.105923 E about 578.81 km radius with relative risk (RR) of 1.57. It surrounds all zones of

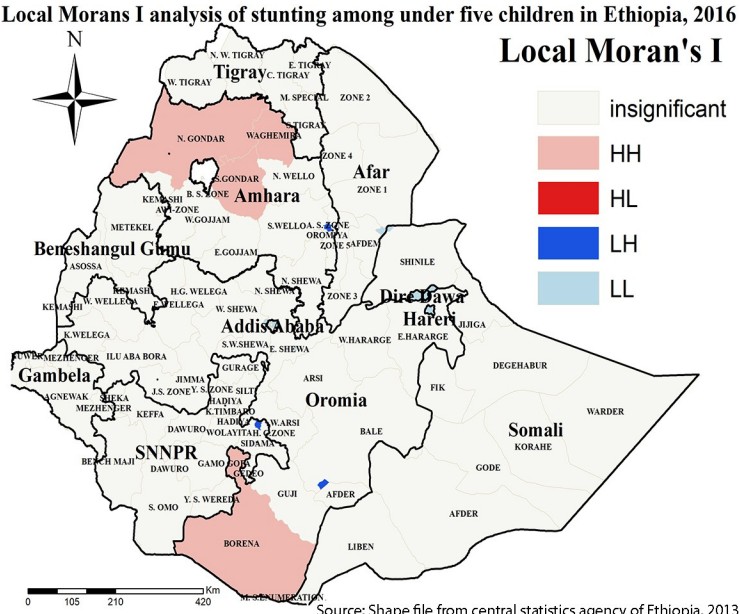

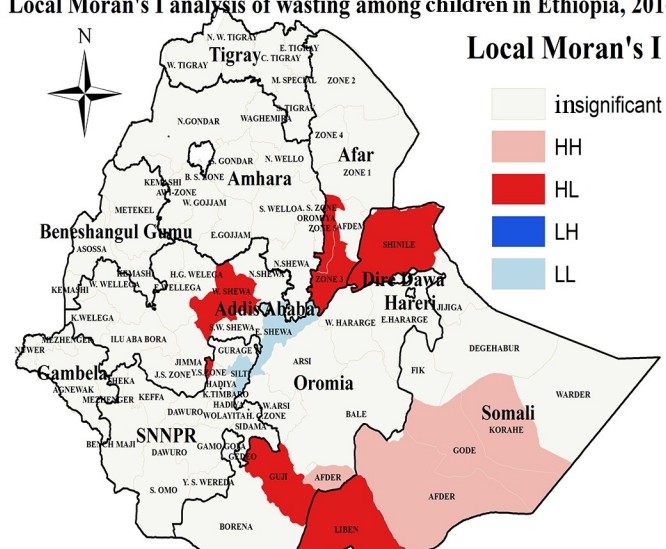

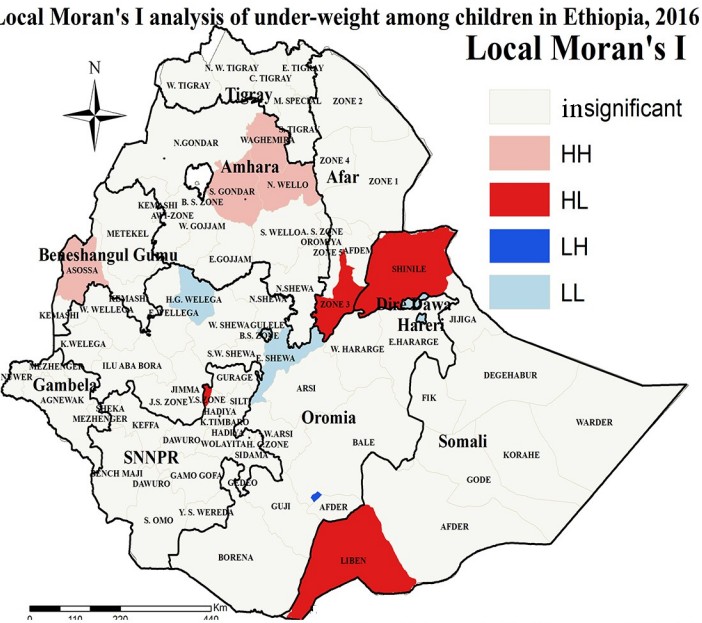

**Fig 6. Cluster and outlier analysis of childhood under-nutrition in Ethiopia.** Local Moran's I analysis result of stunting, wasting, and underweight in Ethiopia from Ethiopian demographic and health survey 2016. The right panel represents for stunting, the left panel represents for wasting, and the right bottom panel represents for underweight. In the figure the damp red color indicates high outlier, the blue color indicates low outlier clustering, the waterish blue color indicates low clustering the fogy color indicates insignificant clustering. The thick black line indicates reginal borders while the thin white black line indicates zonal borders. *Shape file source: (CSA, 2013; https://africaopendata.org/dataset/ethiopia-shapefiles); Own Map output: using ArcGIS 10.7 Software analysis.*

Amhara, Tigray, and Afar regions. In addition, Kelem Wollega, Horo Gudu Wollega, West Shoa, South West Shoa and North Shoa Zones of the Oromia region also contains primary clusters for underweight (Fig 8 and Table 3).

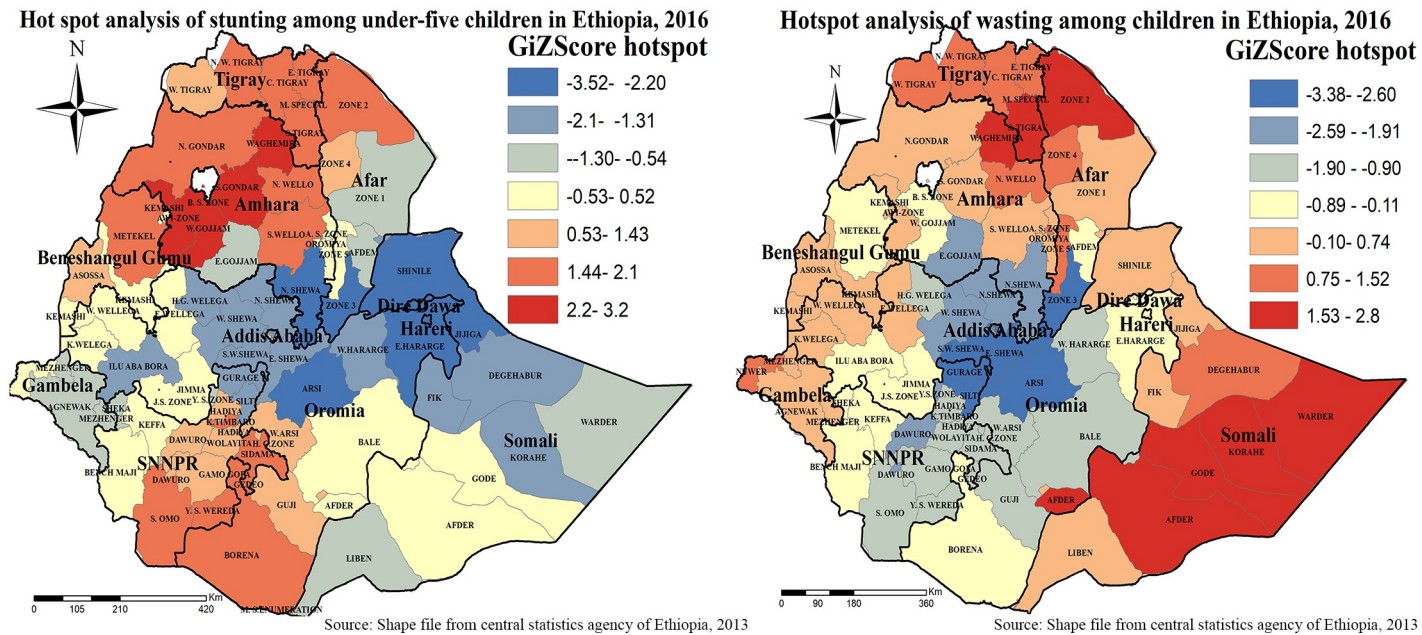

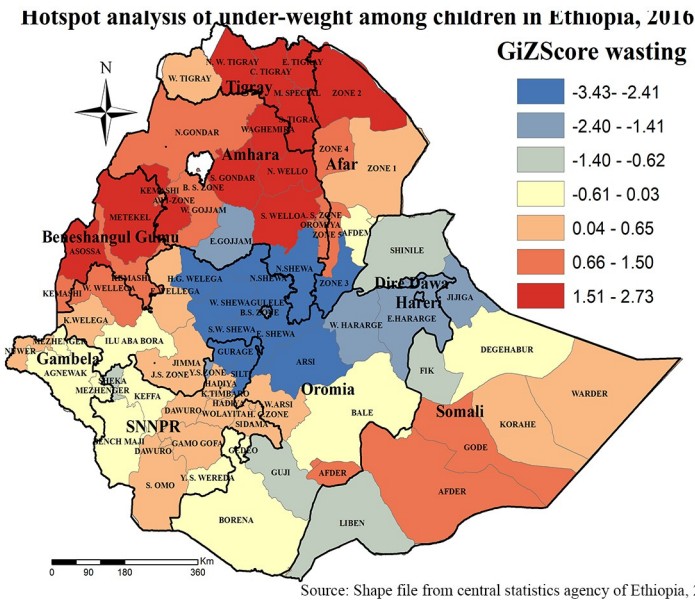

**Fig 7. Hotspot and cold spot zones of childhood under-nutrition in Ethiopia.** Gettis-Ord Gi* local Hotspot analysis result of stunting, wasting and underweight in Ethiopia from Ethiopian demographic and health survey 2016. The right panel represents for stunting, the left panel represents for wasting, and the right bottom panel represents for underweight. In the figure the red color indicates high hotspot zones, the blue color indicates cold spot zones, the whitish yellow color indicates insignificant clustering of wasting, stunting, and underweight. The thick black line indicates reginal borders, while the thin white black line indicates zonal borders. *Shape file source: (CSA, 2013; https://africaopendata.org/dataset/ethiopia-shapefiles); Own Map output: using ArcGIS 10.7 Software analysis.*

## Spatial regression analysis of the determinants of childhood under-nutrition

**Ordinary least square.** After checking spatial regression assumptions for stunting, wasting, and underweight using exploratory regression, an ordinary least square analysis was carried out. Outputs from the spatial regression analysis revealed that, residuals of spatial

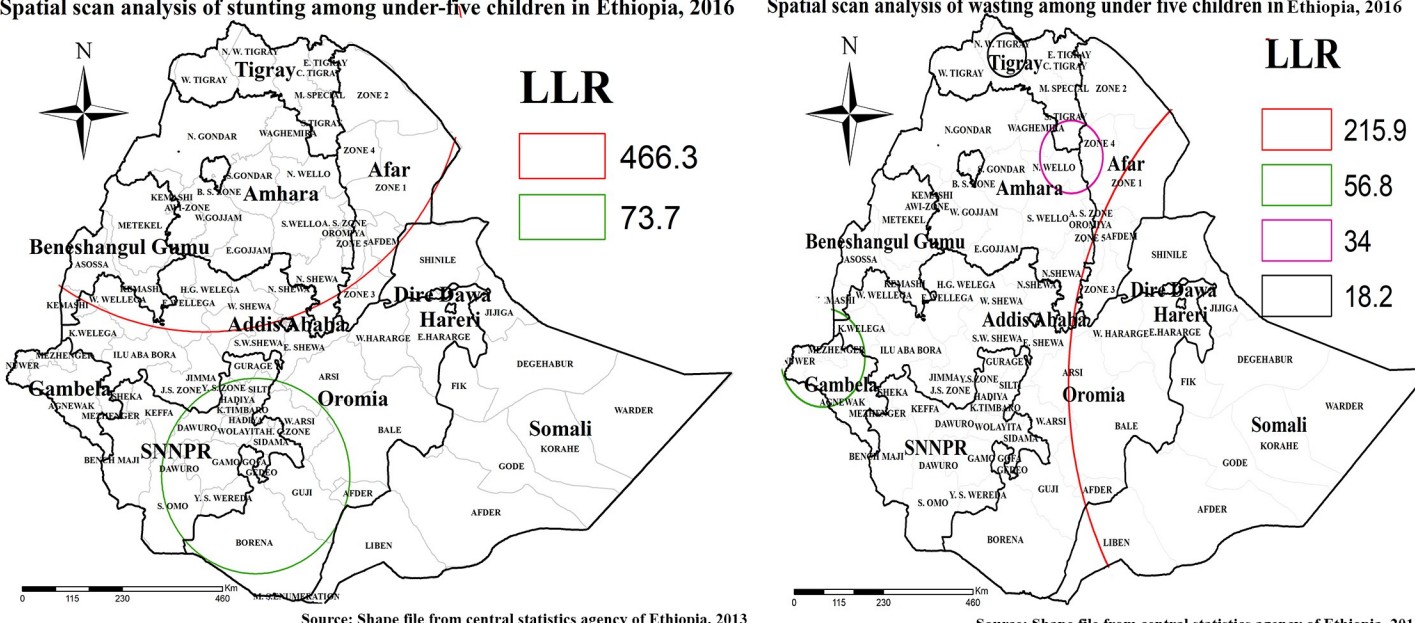

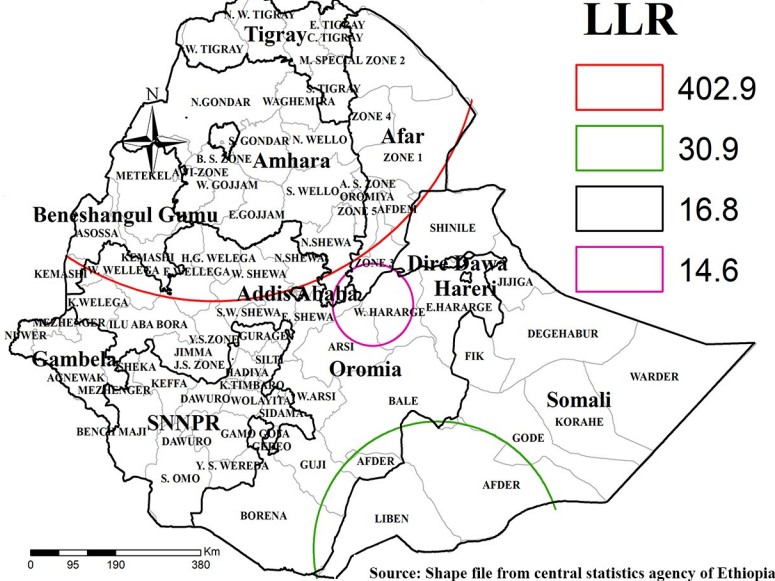

**Fig 8. Spatial clustering of under-nutrition among under-five children in Ethiopia at zonal level.** Spatial scan statics analysis result of wasting, stunting, under-weight in Ethiopia from Ethiopian demographic and health survey. The right panel represents for wasting EDHS, and the left panel represents for stunting EDHS the right bottom panel represents for under EDHS In the figure the red color rings indicates primary spatial window, the rose, black, and green colored ring indicates secondary spatial windows. The foggy line indicates zonal borders while the thick black line indicates regional borders. *Shape file source: (CSA, 2013; https://africaopendata.org/ dataset/ethiopia-shapefiles); Own Map output: using ArcGIS 10.7 Software analysis.*

relationship are uncorrelated (Fig 9, Tables 4–6) and there was no multi-collinearity among explanatory variables (Tables 7–9).

To handle geographical weighted regression, the koenker Bp test in the ordinary least square analysis could be significant, which reveals the difference of coefficients across enumeration areas. In this study, koenker Bp test was found significant hence executing geographically

**Table 3. Significant cluster of under-nutrition in Ethiopia at zonal level.**

| Variable | Detected clusters | Coordinate/radius | Population | Cases | RR | LLR |
|---|---|---|---|---|---|---|
| Underweight | Primary cluster** | (14.033876 N, 37.105923 E)/578.81 km | 23800 | 6908 | 1.57 | 402.9 |
| | Secondary cluster** | (4.006703 N, 41.599743 E)/276.07 km | 800 | 284 | 1.54 | 30.9 |
| | Secondary cluster** | (7.701180 N, 37.486553 E)/0 km | 100 | 50 | 2.15 | 16.8 |
| | Secondary cluster** | (8.757437 N, 40.299442E)/88.72 km | 1205 | 361 | 1.3 | 14.6 |
| Stunting | Primary cluster** | (14.033876 N, 37.105923 E)/578.81 km | 23800 | 9948 | 1.44 | 466.3 |
| | Secondary cluster** | (5.977586 N, 38.145000 E)/214.49 km | 7000 | 2890 | 1.22 | 73.7 |
| Wasting | Primary cluster** | (7.650693 N, 47.007919 E)/819.42 km | 9404 | 1666 | 1.8 | 215.9 |
| | Secondary cluster** | (8.229235 N, 33.886337 E)/107.40 km | 3700 | 625 | 1.56 | 56.8 |
| | Secondary cluster** | (12.161379 N, 39.633205 E)/79.10 km | 2600 | 429 | 1.5 | 34 |
| | Secondary cluster** | (14.157544 N, 38.143852 E)/47.55 km | 900 | 162 | 1.6 | 18.2 |

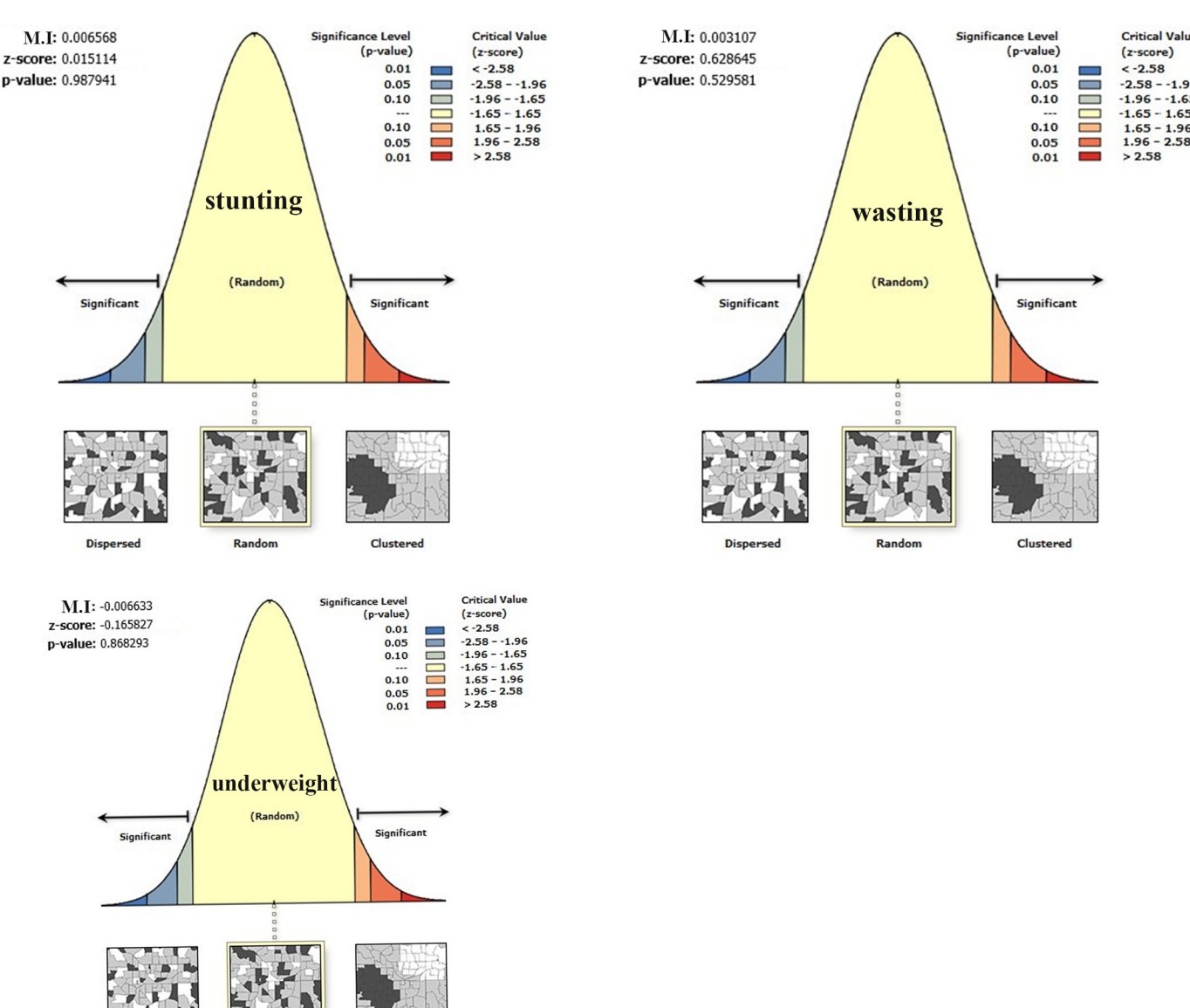

**Fig 9. Spatial autocorrelation of residuals for geographical weighted regression.**

**Table 4. Spatial regression summary result of ordinary least square (global GWR) diagnostics for stunting using 2016 EDHS data.**

| Diagnostics criteria | Magnitude | p-value |
|---|---|---|
| AICc | 5488.44 | |
| R squared | 0.28 | |
| Adjusted R squared | 0.28 | |
| Joint f statistics | 49.9 | 0.0000* |
| Joint wald statistics | 258.48 | 0.0000* |
| Koenker (Bp) statistics | 7.923847 | 0.0056* |
| Jareque-Bera statistics | 41.17 | 0.1173 |

**Table 5. Spatial regression summary result of ordinary least square (global GWR) diagnostics for wasting using 2016 EDHS data.**

| Diagnostics criteria | Magnitude | p-value |
|---|---|---|
| AICc | 5020.45 | |
| R squared | 0.25 | |
| Adjusted R squared | 0.26 | |
| Joint f statistics | 10.08 | 0.0000* |
| Joint wald statistics | 76.42 | 0.0000* |
| Koenker (Bp) statistics | 7.46 | 0.0356* |
| Jareque-Bera statistics | 102.5 | 0.1003 |

**Table 6. Spatial regression summary result of ordinary least square (global GWR) diagnostics for under-weight using 2016 EDHS data.**

| Diagnostics criteria | Magnitude | p-value |
|---|---|---|
| AICc | 5370.16 | |
| R squared | 0.25 | |
| Adjusted R squared | 0.24 | |
| Joint f statistics | 35.42 | 0.0000* |
| Joint wald statistics | 211.26 | 0.0000* |
| Koenker (Bp) statistics | 15.13 | 0.0034* |
| Jareque-Bera statistics | 124.27 | 0.1255 |

**Table 7. Spatial regression summary result of ordinary least square (global GWR) coefficients for stunting using 2016 EDHS data.**

| Variables | Coefficient | Probability | Robust Probability | Vif |
|---|---|---|---|---|
| Intercept | 20.394627 | 0.0000 | 0.0000 | |
| Unimproved toilet | 0.186728 | 0.0000 | 0.0000 | 2.6 |
| Female children | 0.133730 | 0.000923 | 0.006641 | 1 |
| Father has primary education | -0.127514 | 0.002602 | 0.005114 | 1 |
| Father has secondary education | -0.206403 | 0.0000 | 0.0000 | 1.9 |
| Rural residence | -0.030603 | 0.229836 | 0.201706 | 3.1 |

**Table 8. Spatial regression summary result of ordinary least square (global GWR) coefficients for wasting using 2016 EDHS data.**

| Variables | Coefficient | Probability | Robust Probability | Vif |
|---|---|---|---|---|
| Intercept | 5.377181 | 0.0000 | 0.000286 | |
| Unimproved toilet | 0.059517 | 0.006054 | 0.013870 | 2.6 |
| Having 8 family size & above | 0.085499 | 0.000920 | 0.003609 | 1.1 |
| Father with primary education | -0.071297 | 0.013270 | 0.000012 | 1 |
| Mother aged 35–49 | -0.115005 | 0.001641 | 0.000730 | 1.1 |
| Rural residence | 0.005595 | 0.325846 | 0.325412 | 2.6 |

weighted regression is recommended. Accordingly, we have executed geographically weighted regression and determined the local coefficients of each independent variable.

**Ordinary least square analysis for factors associated with under-nutrition.** In ordinary least square analysis households used unimproved toilet facility, father completed primary and secondary education and has female children were factors significantly associated with stunting. Respondents who had used unimproved toilet facility and being a female child increases stunting by 0.186728 and 0.133730 times.

A unit increase for father's primary and secondary education stunting decreases by 0.127514 and 0.206403 times respectively (Table 7).

A unit increase for respondent's l who had used unimproved toilet and having children 8 and above increases wasting by 0.059517 and 0.085499 times. In contrast, a unit increase for father had primary education and mothers aged 35–49 years decreases wasting by 0.071297 and 0.115005 times (Table 8).

Similarly, in ordinary least square analysis unimproved toilet, mother had primary education, father has secondary education, and mothers aged 35–49 years was significant predictors of under-weight. Accordingly, a unit increase for respondent's households who had used unimproved toilet increases underweight by 0.134182 times. In contrast, a unit increase for mother's primary education, father's secondary education, and mother's aged 35–49 years decreases underweight by 0.094737, 0.100582 and 0.158624 times respectively (Table 9).

## Geographically weighted regression of under-nutrition

The result of geographically weighted regression identified different variable coefficients for the identified variables on ordinary least square analysis. For stunting higher coefficients of unimproved toilet of households were detected in all parts of Amhara region, most parts of benshangul-gumz region, western parts of gambela region, western and eastern parts of SNNPR, and western oromia region.

Similarly, higher coefficients for father has primary education were detected in central Tigray region, North Eastern parts of SNNPR and Central Oromia region. higher coefficients

**Table 9. Spatial regression summary result of ordinary least square (global GWR) coefficients for under-weight in Ethiopia.**

| Variables | Coefficient | Probability | Robust Probability | Vif |
|---|---|---|---|---|
| Intercept | 14.548280 | 0.0000 | 0.0000 | |
| Unimproved toilet | 0.134182 | 0.000006 | 0.0000 | 2.6 |
| Mothers aged 35–49 | -0.158624 | 0.000881 | 0.000043 | 1.1 |
| mother has primary education | -0.094737 | 0.000726 | 0.003601 | 1.1 |
| Father has secondary education | -0.100582 | 0.000283 | 0.000314 | 1.9 |
| Rural residence | 0.031254 | 0.185344 | 0.147858 | 3.2 |
| Distance is a big problem | 0.026273 | 0.236677 | 0.253878 | 1.5 |

for father have secondary education was detected in Central Tigray, Eastern parts of SNNPR, most parts of Gambela region and Western Oromia, and higher coefficients for mother's primary education was detected in Central, Eastern and Southern Tigray region, Eastern Amhara region and Northern SNNPR (Fig 10).

For wasting higher coefficients of unimproved toilet of households were detected in eastern Tigray, Eastern Amhara, Eastern parts of SNNPR region and Western parts of Gambella region. Consistently, higher coefficients for father has primary educational level were detected in most parts of Tigray region, Eastern parts of SNNPR region, Harari region and Diredawa city administrative, higher coefficients for mothers aged 35–49 years old were detected in most parts of Tigray region, Eastern parts of Somali region, Harari region and Dire-dawa city administrative (Fig 11).

Similarly, for under-weight higher coefficients of unimproved toilet of households were detected in all regions of the country except Oromia region, higher coefficients for mother had primary education was detected in Tigray region, Gambela region, and Diredawa city administrative. higher coefficients for father have secondary education was detected in Eastern Tigray, central and Eastern Amhara region and Addis Ababa city administrative (Fig 12).

## Discussion

Under-nutrition remains a significant problem in Ethiopia, even efforts have been exerted to reduce the problem. In this study the prevalence of under-nutrition (stunting, wasting and underweight) was 38.30%, 10.10% and 23.54%, respectively. The prevalence of under-nutrition (stunting, wasting and under-weight) was also varying across Ethiopian administrative regions. The burden of under-nutrition is still very high and not evenly distributed in the country. The reports of EDHS also ensure a very stagnant reduction of under-nutrition (stunting, wasting and underweight).

This finding for stunting was lower than studies conducted in Ethiopia [11], in Ghana [9], in Zambia [54], and in Cameroon [55]. But it was higher than studies conducted in Tanzania [56], in Kenya [54], in Ghana [57], and in Senegal [58]. This finding was also in line with studies conducted in different parts of Ethiopia previously [27]. This finding for wasting is lower when compared with studies conducted in Ethiopia [59], in Senegal [58], and Cameroon [55]. But it is in line with studies conducted in Ghana [60], Haromaya Ethiopia [11] and higher than studies conducted in Tanzania [56], Northern Ghana [57], rural Ethiopia [10].

Finding from this study for under-weight is also lower when compared with studies conducted in Ethiopia [61], and Cameroon [55]. But the finding was higher than studies conducted in Ethiopia [62], Tanzania [56], and Northern Ghana [57]. The find was consistent with studies conducted in Ethiopia [11].

The discrepancy might be due to; those pocket studies in Ethiopia might not use adequate representative sample size [63,64]. In addition, it could be due to health system structure and the health policy of Ethiopia, in which different studies conducted in Ethiopia revealed that, the distribution of health care facility and health care professionals across the country is not evenly distributed [65,66].

The finding showed that under-nutrition indicators was clustered spatially at the zonal level. Getis-Ord spatial analysis detected hot spot, cold spot and outliers' zones in Ethiopia. Furthermore, the detection of hot spot zones using Getis-Ord analysis was assessed using sat scan analysis and reports confirmatory findings for the analysis.

In this study the spatial distribution of stunting was clustered at the zonal level. Different studies also support the presence of geographical variation for stunting. A study conducted on geographical variation of stunting found a significant clustering of stunting across Ethiopia

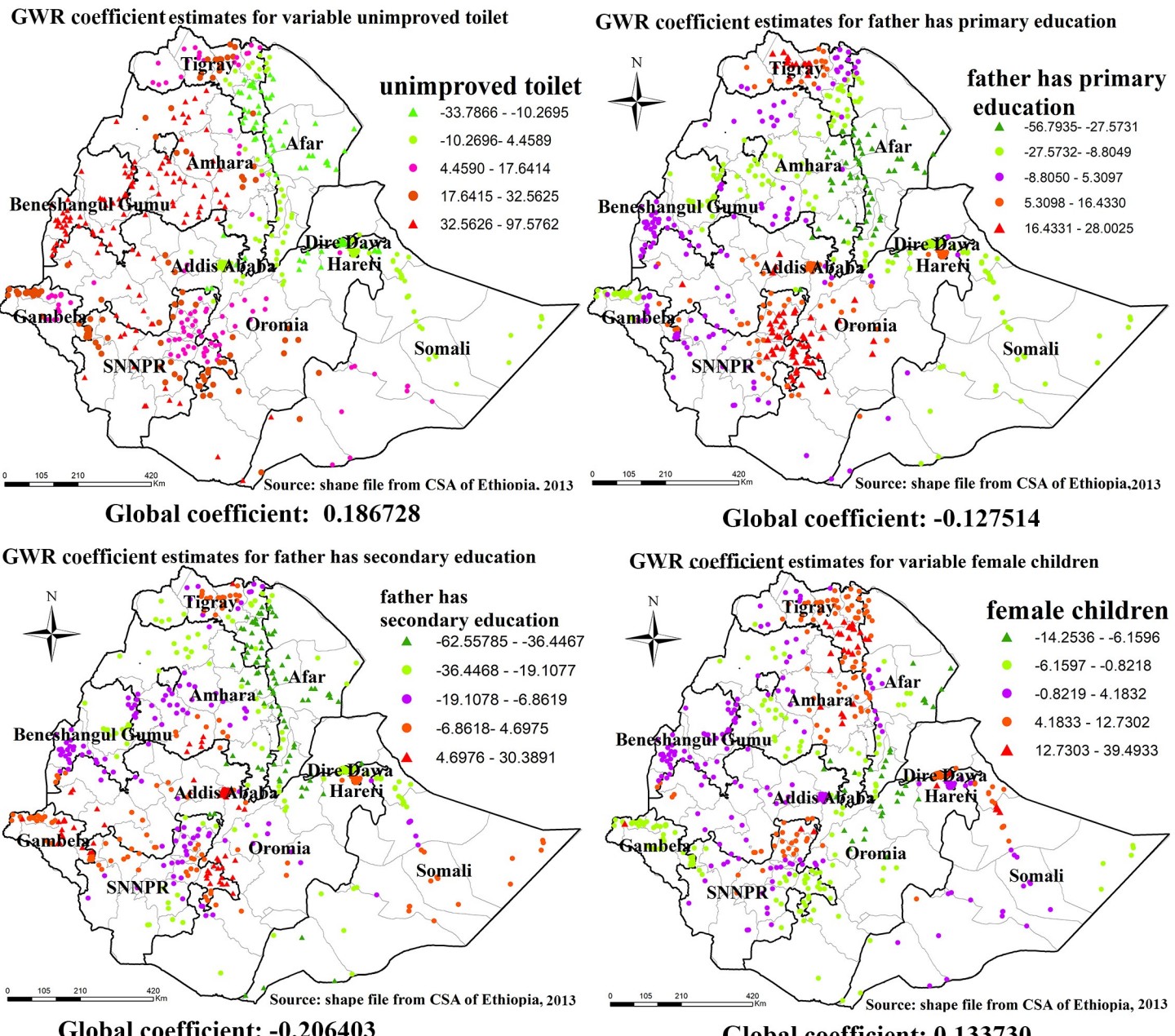

**Fig 10. GWR coefficient estimates for predictors of stunting.** Geographically weighted regression (GWR) analysis result of explanatory variables (unimproved toilet, father has primary education, father has secondary education and female children) with stunting in Ethiopia from Ethiopian demographic and health survey 2016. Each point in the map represents one enumeration area which has a lot of stunting cases. The right panel represents for unimproved toilet, the left panel represents for father has primary education, the right bottom panel represents for father has secondary education and the right bottom panel represents for female children. In the figure the bright red color indicates high magnitude of coefficient estimates, the grean color indicates low magnitude of coefficient estimates, the rose color indicates medium magnitude of coefficient estimates. The thick black line indicates reginal borders while the thin white black line indicates zonal borders. *Shape file source: (CSA, 2013; https://africaopendata.org/dataset/ethiopia-shapefiles); Own Map output: using ArcGIS 10.7 Software analysis.*

zones [28]. The study conducted in India demonstrates a significantly higher value of Moran I (0.65) that suggests a high level of spatial clustering of childhood stunting in zones. Thus, a total of 159 hot spots zones mostly from the central and eastern parts of India [67].

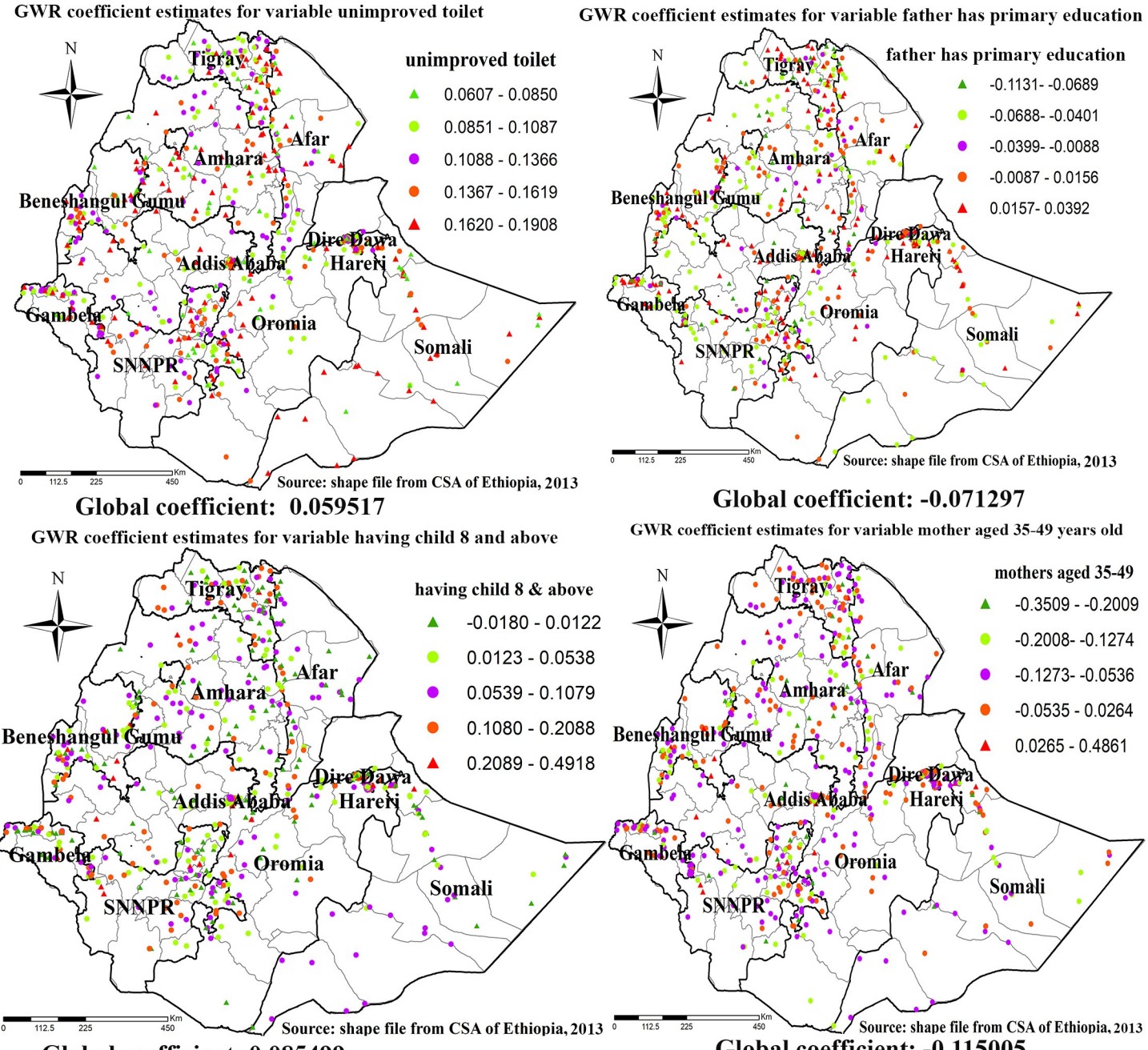

**Fig 11. GWR coefficient estimates for predictors of wasting.** Geographically weighted regression (GWR) analysis result of explanatory variables (unimproved toilet, father has primary education, having children 8 and above and mothers aged 35–49) with wasting in Ethiopia. Each point in the map represents one enumeration area which has a lots of under-weight cases. The right panel represents for unimproved toilet, the left panel represents for father has primary education, the right bottom panel represents for having children 8 and above and the right bottom panel represents for mothers aged 35–49. In the figure the bright red color indicates high magnitude of coefficient estimates, the green color indicates low magnitude of coefficient estimates, the rose color indicates medium magnitude of coefficient estimates. The thick black line indicates reginal borders while the thin white black line indicates zonal borders. *Shape file source: (CSA, 2013; https://africaopendata.org/dataset/ethiopia-shapefiles); Own Map output: using ArcGIS 10.7 Software analysis.*

In this study the spatial distribution of wasting was clustered. Different studies also revealed the presence of geographical variation for wasting. A study conducted in Ethiopia reported that significant clustering of stunting across Ethiopian zones. Thus, Liben, Afder and Shinile

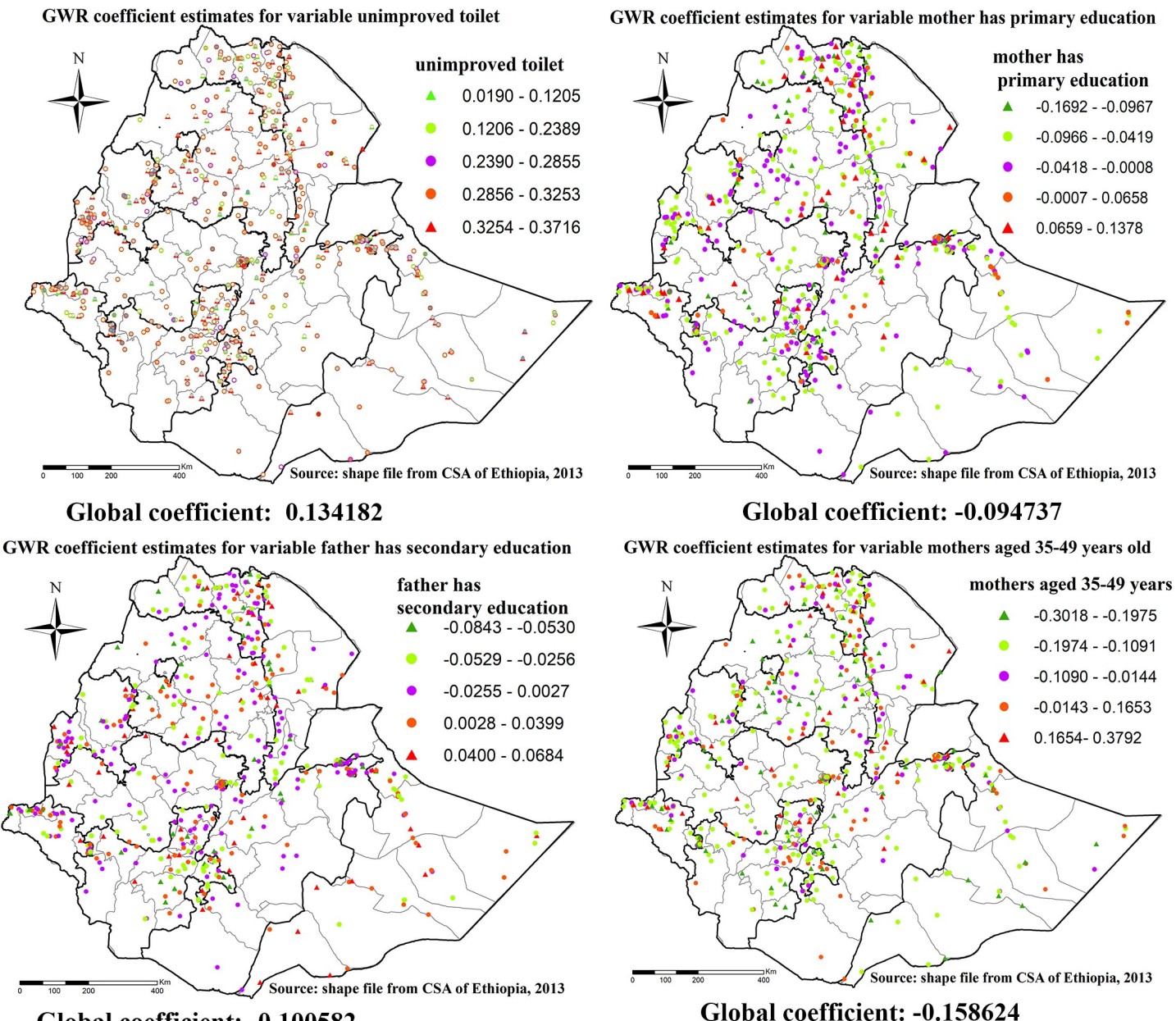

**Fig 12. GWR coefficient estimates for predictors of underweight.** Geographically weighted regression (GWR) analysis result of explanatory variables (unimproved toilet, mothers has primary education, father has secondary education and mothers aged 35–49) with under-weight in Ethiopia. Each point in the map represents one enumeration area which has a lots of under-weight cases. The right panel represents for unimproved toilet, the left panel represents for mothers has primary education, the right bottom panel represents for father has secondary education and the right bottom panel represents for mothers aged 35–49.In the figure the bright red color indicates high magnitude of coefficient estimates, the grean color indicates low magnitude of coefficient estimates, the rose color indicates medium magnitude of coefficient estimates. The thick black line indicates reginal borders while the thin white black line indicates zonal borders. *Shape file source: (CSA, 2013; https:// africaopendata.org/dataset/ethiopia-shapefiles); Own Map output: using ArcGIS 10.7 Software analysis.*

administrative zones from Somali Regional State and Zone 1, Zone 3 and Zone 4 administrative zones from Afar Regional State, Borena, East and West Harargie from Oromiya Regional State were identified as hot spot zones for wasting [29]. A study conducted previously reported that significant clustering of wasting across regions. Thus, Afar and Somali regions were identified as hot spot areas for wasting [27]. A study conducted on spatial clustering of stunting

and wasting in Meskane Mareko District in Gurage Zone found significant clustering of wasting in Diram and Bati Lejano *kebeles* of the district [68].

The current study revealed the spatial distribution of underweight was clustered at Zonal level of Ethiopia. Different studies also revealed the presence of geographical variation for underweight [68]. All administrative zones of Amhara region, Tigray, Afar, Ben. Gumz regional state administrative zones and East Wellega and East Shewa, North Showa administrative zones, from Oromiya Regional State, Liben, Afder, Borena and Gode administrative of Somali regional state were identified as hotspot zones for underweight [29]. A study conducted previously found a significant clustering of underweight across Ethiopia regions. Since Afar and Somali regions were identified as hot spot areas for underweight [27]. A study conducted in India also identified clusters of underweight, 146 districts (23% of all the districts) were observed as of high underweight [67].

The most convenient explanation for this spatial variation of under-nutrition could be as a result of geographical variation in the country which ranges from 4550 meters above sea level to 110 meters below sea level. Consequently, infrastructure differences like: road, electricity, water, the distribution of health facilities, and health care professionals across regions. In addition, there is a difference in culture, socio-demographic characteristics of mothers and fathers, attitude and knowledge difference of the society towards under-nutrition across different regions. Overall this form of difference in Ethiopia could came up with inequalities of under-nutrition indicators across different parts of the country [66,69,70]. This implies that, geographic-based nutritional interventions, mainly mobilizing additional resources could be held to reduce the burden childhood under-nutrition in hot spot areas. In addition, governmental and non-governmental organization working in the identified hot spot areas could be routinely monitored and evaluated nutritional programs in order to improve nutritional status of the children with a special emphasis.

In spatial regression analysis, different statistically significant predictor variables of all forms of under-nutrition were identified. had unimproved toilet, father had primary education; father had a secondary education and has female children were significant predictors for stunting. Similarly, respondents had unimproved toilet, father has primary education, having children 8 and above and mothers aged 35–49 years were significant predictors for wasting.

Further respondents had unimproved toilet facility, mother has primary education, father has secondary education and mothers aged 35–49 years was identified as significant predictors for underweight.

Different studies conducted in Ethiopia and abroad also revealed the existence of considerable significant difference of under-nutrition predictors across a geographical area [24–26,71–73]. The education level of mothers' and the fathers was negatively correlated with under-nutrition indicators (stunting, wasting and underweight) and its coefficients significantly varied across regions of Ethiopia. This might be due to the reason that there is considerable infrastructure difference among regions which in turn inhibits the educational status of the society. Education is a key tool to acknowledge nutritional status of children and to solve problems associated with under-nutrition.

Those who are educated are in a better advance than who are not educated in most occasion. Thus, when education level increase under-nutrition could be decreased. This implies that, Education of parents could be encouraged using different strategies like organizing mothers at village level and assigning community educators, since most of the mothers are aged enough and are not eligible to have formal education. In addition, young mothers could be encouraged to be involved in formal education. Furthermore, the life style of the community could be improved.

In the context of unimproved toilets; on most occasion those households having unimproved toilets could be linked with the poor wealth status of the household and lower education level of the respondents this in turn could be influence the nutritional status of the children. This is because of the reasons that unimproved toilet could increase under-nutrition of the children. This implies that, the government should focus on the construction of independent toilet for each household. Hygiene, sanitation and utilization of household toilets could be improved by frequent education using different agents other than health extension workers. In addition, Hygiene and sanitation as a course could be incorporated in formal education of Ethiopia.

In the context of households have 8 and above members; mothers who have children in the past enable to understand the possible components of adequate nutrition of a child through frequent exposure and experience. In addition, when household had a greater number of members, they could assist in the preparation of foods for their mothers that enable the mother to afford varieties of food for their children easily. This implies that, having greater number of family members could bring the opportunity of caring a child and improve the nutritional status of a child. In the context of mothers aged 35–49 years old; when the age of the mother increases, she could have experiences in caring a child in different occasions. Thus, mothers getting older could acknowledge the nutritional status of their children easily. Since they are well experienced in doing so. This implies that, experience has paramount importance in appreciating and solving a certain problem.

For stunting the analysis of hot spot and geographically weighted regression fit with higher coefficients of unimproved toilet of household's areas identified by GWR. In those areas which has higher coefficients for unimproved toilet of household's also has hot spots of stunting, in those areas which has higher negative coefficients for father has primary and secondary education also has hot spot of stunting.

For wasting the analysis of hot spot and geographically weighted regression fits with higher coefficients of unimproved toilet of household's areas identified by GWR. In those areas which has higher coefficients for unimproved toilet of household's also has hot spots of stunting, in those areas which has higher negative coefficients for fathers has primary education also has hot spot of wasting. Similarly, in areas which have higher negative coefficients for mothers aged 35–49 years old has also hot spot zones for wasting.

For underweight, the analysis of hot spot and geographically weighted regression fit with higher coefficients of unimproved toilet of household's areas identified by GWR. In those areas which has higher coefficients for unimproved toilet of household's also has hot spots of under-weight, in those areas which has higher negative coefficients for fathers and mother has primary education also has hot spot of under-weight.

Over all the analysis of hot spot areas using Getis-Ord spatial auto correlation cluster and outlier and spatial scan statistics fits with the analysis of spatial regression in somewhere and vary also somewhere. This study supports the existing knowledge on the influence of infrastructure coverage, and geographic features on under-nutrition across the country. This study contributes to identify specific hot spot zones throughout the country and factors significantly affect under-nutrition, which is very important for intervention. Even though there is documented influence of factors like infrastructure coverage and geographic features it is difficult to understand specific hot spot zones for under-nutrition in which this study could brought a solution. The findings of the present study are crucial for zonal level planning for child health. Regional pattern and clustering of indicators used in the study suggest a need to strengthen and continue zonal focused programs.

### Strength and limitation of the study

The study findings can be generalized to all under-five children in Ethiopia. Moreover, the use of Geographic Information System (GIS) and Sat Scan statistical tests helped to detect similar and statistically significant high-risk clusters/hotspots of under-nutrition. Additionally, use of geographic weighted regression analysis helps to show the real impact of predictors at each specific geographic area. Furthermore, this study had used geographically weighted regression analysis that could enables to determine local coefficients a step advance from ordinary least square analysis. In addition, geographically weighted regression analysis improves model performance by employing a spatial weighted function and addresses spatial heterogeneity and also has an advantage of having less biased predicted coefficients compared to spatial lag and spatial error models. Even though spatial lag and spatial error models enable to addresses both spatial heterogeneity and spatial homogeneity, spatial heterogeneity and spatial homogeneity are unique, not mutually exclusive, features of spatial data. Therefore, this model might bring biased coefficients.

As to limitation, location data values were shifted 1-2km for urban and 5km for rural areas for data confidentiality reasons since it didn't show exact case locations. The data used for analysis were not taking into account seasonal variations of under-nutrition. In addition, the study did not do an adjustment for covariates during estimation of the spatial epidemiology of child malnutrition. The missed data may also affect the true estimates of the analysis. Furthermore, there may be false inclusion and exclusion of SaT Scan clusters.

## Conclusion

Our study showed that the distribution of under nutrition among under-five children was clustered at zonal level in Ethiopia. It had a geographical variation across regions of Ethiopia. Hot spot zones for stunting were detected in West Gojam, Awi, South Gondar, and Waghmira zones). Similarly, Somali region (Afder, Gode, Korahe, Warder Zones), Afar region (Zone 2), Tigray region (Southern zone), and Amhara region (Waghmira zones) was detected as hot spot zones for wasting. Furthermore, hot spot zones for under-weight was detected in Amhara region (South Wollo, North Wollo, Awi, South Gondar, and Waghmira zones), Afar region (Zone 2), Tigray region (Eastern zone, North Western zone, Central zone, Southern zone, and Mekele special Zones), and Benshangul region (Metekel and Assosa Zones).

In spatial regression analysis, different statistically significant predictor variables of all forms of under-nutrition were identified. Had unimproved toilet, father with primary and secondary education, and has female children were significant predictors of stunting. Consistently, has unimproved toilet, father has primary education, having children 8 and above, and mothers aged 35–49 years were also significant factors of wasting. In addition, has unimproved toilet, mother has primary education, father has secondary education, and mothers aged 35–49 years were significant predictors of underweight.

Thus, geographic based nutritional interventions mainly mobilizing additional resources could be held to reduce the burden childhood under-nutrition in hot spot areas. In addition, improving sanitation and hygiene practice, improving the life style of the community, and promotion of parent education in the identified hot spot zones for under-nutrition should be more emphasized.

## Supporting information

**S1 File. Conceptual frame work of the study.**
(PDF)

**S2 File. Variance inflation factor for stunting.**
(PDF)

**S3 File. Variance inflation factor for wasting.**
(PDF)

**S4 File. Variance inflation factor for underweight.**
(PDF)

**S5 File. Letter of approval to use DHS data.**
(PDF)

## Acknowledgments

We are grateful to the MEASURE DHS program that provides permission with data access authorization so as to enable us to conduct the study.

## Author Contributions

**Conceptualization:** Amare Muche, Mequannent Sharew Melaku, Erkihun Tadesse Amsalu, Metadel Adane.

**Data curation:** Amare Muche, Mequannent Sharew Melaku, Erkihun Tadesse Amsalu, Metadel Adane.

**Formal analysis:** Amare Muche, Mequannent Sharew Melaku, Erkihun Tadesse Amsalu, Metadel Adane.

**Investigation:** Amare Muche, Mequannent Sharew Melaku, Erkihun Tadesse Amsalu, Metadel Adane.

**Methodology:** Amare Muche, Mequannent Sharew Melaku, Erkihun Tadesse Amsalu, Metadel Adane.

**Project administration:** Amare Muche, Mequannent Sharew Melaku, Erkihun Tadesse Amsalu, Metadel Adane.

**Resources:** Amare Muche, Mequannent Sharew Melaku, Erkihun Tadesse Amsalu, Metadel Adane.

**Software:** Amare Muche, Mequannent Sharew Melaku, Erkihun Tadesse Amsalu, Metadel Adane.

**Validation:** Amare Muche, Mequannent Sharew Melaku, Erkihun Tadesse Amsalu, Metadel Adane.

**Visualization:** Amare Muche, Mequannent Sharew Melaku, Erkihun Tadesse Amsalu, Metadel Adane.

**Writing – original draft:** Amare Muche, Mequannent Sharew Melaku, Erkihun Tadesse Amsalu, Metadel Adane.

**Writing – review & editing:** Amare Muche, Mequannent Sharew Melaku, Erkihun Tadesse Amsalu, Metadel Adane.

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
