## [Decision Letter · Decision Letter 0]

12 Nov 2020

PONE-D-20-27277

Geographical clustering of under-nutrition and its predictors among under-five children in Ethiopia: Geographically weighted regression Analysis

PLOS ONE

Dear Dr. sharew,

Thank you for submitting your manuscript to PLOS ONE. After careful consideration, we feel that it has merit but does not fully meet PLOS ONE’s publication criteria as it currently stands. Therefore, we invite you to submit a revised version of the manuscript that addresses the points raised during the review process.

Kindly look at the comments of all four reviewers and revise and resubmit your paper. In particular, I invite your attention to the second reviewer comments. While submitting the revised version, please format the paper according PLOS One Stylistic guidelines. 

We look forward to receiving your revised manuscript.

Kind regards,

Srinivas Goli, Ph.D.

Academic Editor

PLOS ONE

Journal Requirements:

3. We note that Figures 5, 7, 8, 9, 10, 11 and 12 in your submission contain map images which may be copyrighted.

a. You may seek permission from the original copyright holder of Figures 5, 7, 8, 9, 10, 11 and 12 to publish the content specifically under the CC BY 4.0 license. 

Additional Editor Comments:

Kindly look at the comments of all four reviewers and revise and resubmit your paper. In particular, I invite your attention to the second reviewer comments. While submitting the revised version, please format the paper according PLOS One Stylistic guidelines.

Reviewers' comments:

Reviewer's Responses to Questions

**Comments to the Author**

1. Is the manuscript technically sound, and do the data support the conclusions?

Reviewer #1: Partly

Reviewer #2: Yes

Reviewer #3: Yes

Reviewer #4: Yes

2. Has the statistical analysis been performed appropriately and rigorously? 

Reviewer #1: Yes

Reviewer #2: Yes

Reviewer #3: Yes

Reviewer #4: Yes

3. Have the authors made all data underlying the findings in their manuscript fully available?

Reviewer #1: Yes

Reviewer #2: Yes

Reviewer #3: Yes

Reviewer #4: Yes

4. Is the manuscript presented in an intelligible fashion and written in standard English?

Reviewer #1: Yes

Reviewer #2: Yes

Reviewer #3: Yes

Reviewer #4: Yes

5. Review Comments to the Author

Reviewer #1: Reviewer’s comments

Full title: Geographical clustering of under-nutrition and its predictors among under-five children in Ethiopia: Geographically weighted regression Analysis

Manuscript number: PONE-D-20-27277

Corresponding author: Mequannent Melaku sharew, Mph in health informatics Wollo University Desse, ETHIOPIA

Comments and questions

I appreciate the authors for addressing such highly contextually and culturally affected public health issues. It so concerning that under-nutrition is continued to be a public health problem in Ethiopia. Therefore, evidence regarding the distribution of under-nutrition with its respective determinants is important for decision-making in dealing with its prevention and control program. The paper is well organized and presented. The study presents the findings of original research in the area of public health. And the results have not been published elsewhere. Having adequately performed relevant analyses with sufficient detail; it presents coherent and data-oriented conclusions. The article is presented in an acceptable level English language standard and reporting guideline.

Additional comments and questions

The author would respond to some of the following questions and comments.

1. Title: “Geographical clustering of under-nutrition and its predictors among under-five children in Ethiopia: Geographically weighted regression Analysis”. Since the data for this study comes from the DHS source; it’s good to indicate it in the title. So, the author would restructure the title as…; “Using Geographically weighted regression Analysis to cluster under-nutrition and its predictors among under-five children in Ethiopia: Evidence from demographic and health survey”.

2. Abstract: Line 30: Methods: in the method section of the abstract; the author should indicate that the data comes from DHS. Indicating it is very important as it adds quality to the paper (data from a large population) and attracts the readers at a first impression.

3. Introduction: page: Line 99-10: “Geographical variation of under-nutrition can delay control and elimination of malnutrition which is the underline cause of most childhood disease that in turn brings a high proportion of under-five mortality.” This sentence is not clear or it seems incomplete. It would good for the author to rephrase it?

4. Methods and materials: Study setting: Line 110: the author would briefly discuss information about study settings or Ethiopia. E.g about the nine regional states of Ethiopia and other relevant information to under-nutrition such as urbanization, economic and agricultural practices, etc.

5. Discussion: Line 374 and the Implications of the study: Line 510: It gives more sound meaning if the author could combine discussion points with implications. The author tried to compare the findings of the study with other previous pieces of evidence and discuss discrepancies as well. That’s good. But it would be more informative if the author could also discuss the implications (policy, methodological and practical implications) for each point of discussion and explanations. in so doing, the author would remove the section “Implications of the study in Line 510”.

6. Question: It’s clear that the distribution of under-nutrition in Ethiopia was indicated in DHS 2016. And the determinants as well. My question is; what specific value (scientific or methodological) did your study added to the existing body of evidence? Would you please indicate it clearly in your discussions? Keep in mind my comments in #5 above.

Reviewer #2: The manuscript written in an organized and impressive way. There are minor comments I attached and author has to come up with justification for the comments. I attached the comments in word format for details

Reviewer #3: The author need to address the all the comments properly.

The article can be a good addition in the field of spatial demography provided that author substantially revise the manuscript as per the suggestions.

Reviewer #4: 1. In the abstract part line 31, says that a stratified two stage cluster sampling was used for to include clusters. however, the stratification and stages of sampling has not available in the manuscript. So it is better you will add it in your manuscript.

2. Your manuscript have no page numbers. You have to add it.

6. PLOS authors have the option to publish the peer review history of their article (what does this mean?). If published, this will include your full peer review and any attached files.

Reviewer #1: **Yes: **Fira Abamecha

Reviewer #2: No

Reviewer #3: No

Reviewer #4: No

---

## [Author Response · Author response to Decision Letter 0]

27 Dec 2020

dear editors, this are some of my answers for your comment

Thank you, dear editor, for the comment. 

We have written the manuscript according to PLOSE ONE format using the guideline. 

We have revised the language usage, spelling, and grammar of the manuscript.

We have removed the ethics statement that found in the declaration section.

All the figures you noted (5, 7, 8, 9, 10, 11 and 12) was produced using shape file from open Africa website (https://africaopendata.org/dataset/ethiopia-shapefiles) which freely available website and enumeration area shape file from DHS website(https://dhsprogram.com/data/dataset_admin/login_main.cfm?CFID=1242368&CFTOKEN=c892a8da9855f981-8D71EDAA-BF68-5950-1D6BA7F0BB37D5E8) and the we have produced the figures by ArcGIS version 10.7 software. Permission letter to use the Ethiopian enumeration area shape file and non-spatial data is attached as supplementary information(S5). For this reason, we have incorporated the source of Ethiopia shape file URL in each figure.

---

## [Decision Letter · Decision Letter 1]

22 Feb 2021

Using geographically weighted regression Analysis to cluster under-nutrition and its predictors among under-five children in Ethiopia: Evidence from demographic and health survey

PONE-D-20-27277R1

Dear Dr. sharew,

We’re pleased to inform you that your manuscript has been judged scientifically suitable for publication and will be formally accepted for publication once it meets all outstanding technical requirements.

Kind regards,

Srinivas Goli, Ph.D.

Academic Editor

PLOS ONE

Additional Editor Comments (optional):

All comments have been addressed

Reviewers' comments:

Reviewer's Responses to Questions

**Comments to the Author**

1. If the authors have adequately addressed your comments raised in a previous round of review and you feel that this manuscript is now acceptable for publication, you may indicate that here to bypass the “Comments to the Author” section, enter your conflict of interest statement in the “Confidential to Editor” section, and submit your "Accept" recommendation.

Reviewer #2: All comments have been addressed

Reviewer #4: All comments have been addressed

2. Is the manuscript technically sound, and do the data support the conclusions?

Reviewer #2: Yes

Reviewer #4: Yes

3. Has the statistical analysis been performed appropriately and rigorously? 

Reviewer #2: Yes

Reviewer #4: Yes

4. Have the authors made all data underlying the findings in their manuscript fully available?

Reviewer #2: Yes

Reviewer #4: Yes

5. Is the manuscript presented in an intelligible fashion and written in standard English?

Reviewer #2: Yes

Reviewer #4: Yes

6. Review Comments to the Author

Reviewer #2: Thank you Authors. The comments I raised were adequately addressed and in a coherent way. The research is well edited and addressed issues I raised in the previous review process. The only point I have is that figures given are too many and if there is a way to shorten it , the paper will be clear for the readers of the article.

Reviewer #4: (No Response)

7. PLOS authors have the option to publish the peer review history of their article (what does this mean?). If published, this will include your full peer review and any attached files.

Reviewer #2: **Yes: **Adem Abdulkadir Abdi

Reviewer #4: No

---

## [Editor Report · Acceptance letter]

7 Apr 2021

PONE-D-20-27277R1 

Using geographically weighted regression analysis to cluster under-nutrition and its predictors among under-five children in Ethiopia: Evidence from demographic and health survey 

Dear Dr. Melaku:

I'm pleased to inform you that your manuscript has been deemed suitable for publication in PLOS ONE. Congratulations! Your manuscript is now with our production department. 

Kind regards, 

on behalf of

Dr. Srinivas Goli 

Academic Editor

PLOS ONE